# The spatial RNA integrity number assay for in situ evaluation of transcriptome quality

Linda Kvastad[1], Konstantin Carlberg[1,8], Ludvig Larsson[1,8], Eva Gracia Villacampa[1,8], Alexander Stuckey[1], Linnea Stenbeck[1], Annelie Mollbrink[1], Margherita Zamboni[2], Jens Peter Magnusson[2,7], Elisa Basmaci[3,4], Alia Shamikh[3,4], Gabriela Prochazka[3,4], Anna-Lena Schaupp[5], Åke Borg[6], Lars Fugger[5], Monica Nistér[3,4] & Joakim Lundeberg[1✉]

The RNA integrity number (RIN) is a frequently used quality metric to assess the completeness of rRNA, as a proxy for the corresponding mRNA in a tissue. Current methods operate at bulk resolution and provide a single average estimate for the whole sample. Spatial transcriptomics technologies have emerged and shown their value by placing gene expression into a tissue context, resulting in transcriptional information from all tissue regions. Thus, the ability to estimate RNA quality in situ has become of utmost importance to overcome the limitation with a bulk rRNA measurement. Here we show a new tool, the spatial RNA integrity number (sRIN) assay, to assess the rRNA completeness in a tissue wide manner at cellular resolution. We demonstrate the use of sRIN to identify spatial variation in tissue quality prior to more comprehensive spatial transcriptomics workflows.

[1] Science for Life Laboratory, KTH - Royal Institute of Technology (KTH), SE-171 65, Solna, Sweden. [2] Department of Cell and Molecular Biology, Karolinska Institutet, Stockholm, Sweden. [3] Department of Oncology-Pathology, Karolinska Institutet, Stockholm, Sweden. [4] Department of Clinical Pathology and Cytology, Karolinska University Hospital, Stockholm, Sweden. [5] Nuffield Department of Clinical Neurosciences, MRC Human Immunology Unit, Weatherall Institute of Molecular Medicine, John Radcliffe Hospital University of Oxford, Oxford Centre for Neuroinflammation, Oxford, UK. [6] Division of Oncology and Pathology, Department of Clinical Sciences, Lund Lund University, Lund, Sweden. [7] Present address: Bioengineering Department, Stanford University, Stanford, USA. [8] These authors contributed equally: Konstantin Carlberg, Ludvig Larsson, Eva Gracia Villacampa. ✉email: joakim.lundeberg@scilifelab.se

The integrity of RNA is often an independent and essential quality measure when selecting samples for downstream transcriptome studies. A sample's RNA integrity number (RIN) is often defined based on the relative abundance of full-length transcripts[1]. High-quality samples consist predominantly of full-length transcripts while low-quality samples contain mostly short fragmented transcripts. The degree of fragmentation is usually assessed by gel electrophoresis after extraction of total RNA from the investigated tissue[2,3]. Because messenger RNA (mRNA) is expressed in low quantities, RNA integrity in eukaryotic cells is commonly determined by quantifying the more abundant 18S and 28S ribosomal RNA (rRNA) transcripts[1]. Previous studies have also suggested that the RIN estimates should be considered when comparing gene expression data between bulk samples[4,5] as well as within a sample[6].

Recently, several novel spatial methods have been developed to study mRNA transcripts in situ[7–12], allowing us to visualize the spatial distribution of the mRNA transcriptome. Frequently, a bulk RNA analysis is used as a pre-screen for such spatial methods without considering the RNA integrity in situ. This is of particularly importance for clinical samples that are collected at highly variable experimental and tissue conditions and/or are too scarce for standard extraction of RNA. Here we present, to our knowledge, a novel method capable of measuring a spatial RNA integrity number (sRIN) in situ from a single tissue section suitable for such clinical samples.

## Results

**The sRIN assay.** To retrieve the spatial RNA information, we designed an assay based on placing single cell layer tissue sections on a glass slide with pre-printed DNA oligonucleotides complementary to the 18S rRNA, each sRIN slide can contain up to 16 fully coated capture areas (7.5 × 7.5 mm). The tissue section is non-covalently fixed, permeabilized, stained with hematoxylin and eosin (HE) for imaging, and incubated overnight to allow the synthesis of transcripts complementary-to-18S-rRNA (c18RNA). The tissue and rRNA templates are subsequently removed, leaving behind a single-stranded c18RNA footprint (Fig. 1a). Finally, fluorescently labeled oligonucleotide probes are sequentially and rapidly hybridized at multiple sites on the c18RNA (Fig. 1b) and

the spatial distribution of RNA integrity is estimated from that of the 18S rRNA (Fig. 1c), which is visualized as a heat map covering the entire tissue section. By comparing this sRIN heat map (Fig. 1d) to images of the tissue section acquired after staining with HE (Fig. 1e), one can determine the distribution of spatial RNA integrity at cellular resolution. As shown here (Fig. 1f, g) we find high sRIN values in both cell dense and cell sparse areas. No signal or low signal is believed to represent lack of cells or low fraction of cells, the latter being more affected by the partitioning of cell structures during cryosectioning.

The gene body coverage of the generated c18RNA was estimated by qPCR (Supplementary Fig. 1), using primer pairs in which one primer sequence was identical to one of the four probes (P1–P4) used in the sRIN assay (Supplementary Table 1). We observed full-length c18RNA gene body coverage with a slight 3′ end bias.

To determine the robustness and durability of the generated c18RNA footprint, we performed several rounds of hybridization using a mouse olfactory bulb tissue sample, alternating between a surface probe and P1 conjugated with Cy5 and Cy3, respectively. After scanning each image and recording its fluorescence signal (measured in fluorescence units, FU), the probe was stripped off and another probe was hybridized. The results demonstrates reproducible sequential probe hybridizations (Supplementary Fig. 2). The c18RNA footprint was also tested over time, revealing that a FU signal was detectable even after storing for 6 months (Supplementary Fig. 3). To evaluate the hybridization efficiency of the four probes (P1–P4) in the sRIN assay, we used capture areas printed with complementary sequences (Supplementary Table 1). The FU measurements for each probe were observed to have similar hybridization efficiencies (Supplementary Fig. 4).

To mimic RNA fragmentation we used total RNA, performed magnesium-based degradation, and mixed samples to represent different levels of RNA degradation (Supplementary Fig. 5a–e). These mixtures were then compared in solution (RIN) and on the slide (sRIN) and this showed that the two measures were similar, having a Pearson correlation coefficient of 0.87 (Supplementary Fig. 5f). We also observed that when the samples with high and low RIN values (9 and 2.9, respectively) were combined to simulate heterogeneous degradation patterns in a ratio of 1:4 (v/v), the new combined sample had a RIN value of 5.6, which

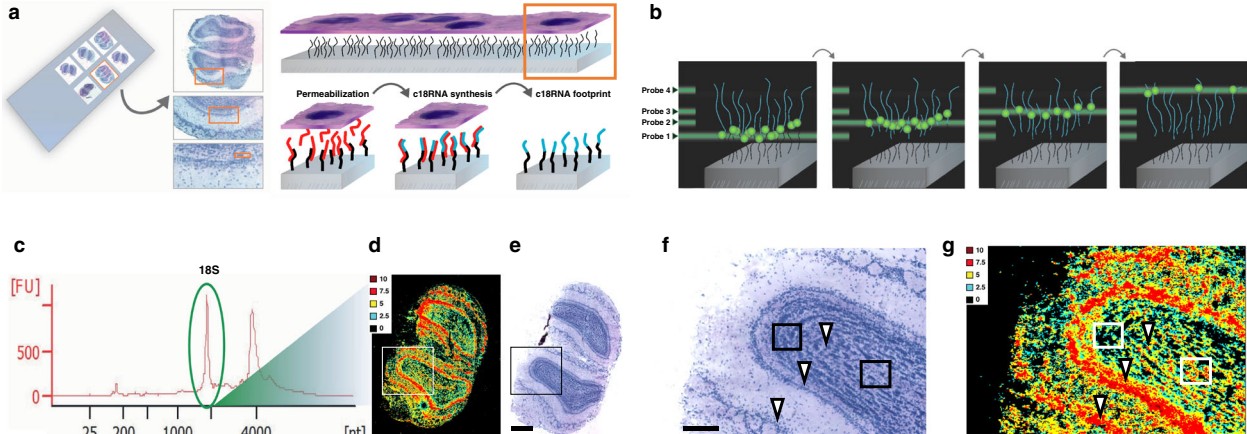

**Fig. 1 The spatial RNA Integrity Number (sRIN) assay. a** Schematics of an one cell layer thick tissue section placed on a pre-printed fully coated capture area for 18S rRNA, leading to the formation of a c18RNA-footprint. **b** Schematic depiction of the sequential probe hybridization. **c** Total RNA gel electropherogram of a mouse olfactory bulb tissue sample together with **d** a heat map visualization of the section's spatial RNA integrity number (sRIN) distribution based on its c18RNA footprint (n = 4, technical replicates) and **e** a HE image of the same section (scale bar: 600 µm), boxes enclose regions shown in close-up. **f** Close-up of HE image (scale bar: 200 µm) and **g** sRIN heat map, arrows marks cells from different areas in the tissue with high sRIN values, the left box depicts a group of cells where cell drop-out has occurred compared to the right box that depicts a group of cells recording high sRIN values.

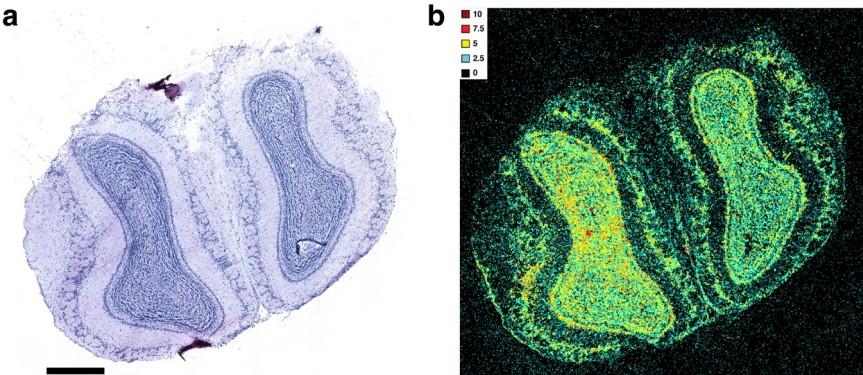

**Fig. 2 The sRIN assay measuring short fragments in situ. a** HE image (scale bar: 600 μm) and **b** sRIN heat map of the same mouse olfactory bulb displayed in Fig. 1d–g, generated from a freshly frozen tissue section after brief formalin (10%) fixation to simulate fragmentation by cross-linking ($n = 4$, technical replicates).

was near the mean values for the original samples (Supplementary Fig. 5a–c). Further analysis demonstrate the consistency with the sRIN assay's 18S rRNA specificity (Supplementary Fig. 6).

**Measuring short fragments in situ with the sRIN assay**. To further investigate the ability of the sRIN assay to measure fragment lengths in situ, we modified the sRIN protocol by performing a brief formalin (10%) fixation to introduce cross-links prior to c18RNA synthesis, generating shorter c18RNA fragments (Fig. 2). The resulting sRIN heat map revealed that the c18RNA lengths (indicated by the sRIN values) in the absence of cross-linking (Fig. 1d, g) differed markedly from those in the cross-linked section (Fig. 2b). Importantly, this fixed mouse brain sample also demonstrated an even distribution of fluorescent signal, representing an optimal handling of tissue, with balanced signals for the multi-layered cellular architecture of the mouse olfactory bulb from surface to center view. This may suggest that observed variable sRIN values for clinical samples could be linked to consistency issues in the preanalytical steps of sample handling.

**The sRIN assay applied to well-preserved clinical samples**. To validate the sRIN assay in tissue sections, we generated sRIN heat maps (Fig. 3a–l) from two selected breast cancer specimens[13] considered to be of good total RNA quality (RIN ≥ 7.5), well-preserved morphology and with accompanying spatial transcriptomics[7,14] (ST) data (Supplementary Fig. 7). The heat maps revealed that strong sRIN signals colocalized with the tissue morphology, whereas signals from the background was, as expected, either weak or non-existent. We observed that one sample exhibited very high and even sRIN values (10) throughout the tissue section (Fig. 3a–d), while another exhibited slightly variable but high sRIN values (7.5–10) (Fig. 3e–l). We also found that very high sRIN values (10) could be observed in areas of both high and low cell density (Fig. 3k–l). In addition, a human post-mortem brain was evaluated for its sRIN heterogeneity, representing a challenging sample type in terms of RNA stability. Figure 3m, n depicts a sRIN assay of gray matter (GM) with consistently high sRIN values (10) and the assay also demonstrates that the RNA integrity of single cells can be estimated. We also applied the sRIN assay to sparser cell white matter (WM) regions, with bulk RIN of similar RNA quality compared to GM regions, as expected observing less number of cells being represented in sRIN from WM compared to GM (Supplementary Fig. 8).

**Applied sRIN assay to a less well-preserved clinical sample**. We applied the sRIN assay on a malignant childhood brain tumor for evaluation of its transcriptome quality (Fig. 4). The generated sRIN data (see Fig. 4a, b and Supplementary Fig. 9) was compared to ST performed on a nearby section. Overall we found that both the spatial distribution of mRNA molecules (Fig. 4c, d) and the exon/intergenic mapping rate (Supplementary Fig. 10a, b) correlated with the sRIN distribution (Fig. 4a, b). Additionally, by analyzing the ST results from the serial tissue sections located both before and after the section used in the sRIN analysis, we found that the spatial RNA quality distribution observed in the sRIN analysis was representative of that in the wider 3D sample space (Supplementary Fig. 9).

From the HE staining of the tumor (Fig. 4), we observed deep blue areas that represented intact cell rich tumor, light purple areas where the tumor was necrotic and dark purple representing patches of calcifications. We also found areas with cracks located mostly around the tissue edges. These cracks were most likely created during the freezing procedure of the sample specimen. By comparing the HE staining with the sRIN heat map we observed larger regions of intact cell rich tumor with very high (10) and high (7.5) sRIN values. In contrast, no sRIN values from the necrotic regions (Fig. 4e, f). We also observed smaller tumor areas with similar overall HE staining pattern that varied greatly in measured sRIN values (Fig. 4g, h). As reference, bulk RIN analyses of the investigated tissue samples show average high RIN values (≥7.5) (Supplementary Table 2) demonstrating the value of a more in-depth in situ analysis as shown here.

## Discussion
The collection of ST methods is increasing and the quality of tissue sections is of paramount importance for successful interpretation. However, the quality of mRNA in clinical samples often vary considerably due to sampling, handling, and storage conditions, and are frequently only available in minute amounts for downstream molecular analysis.

This paper presents an unbiased assay for in situ spatial transcriptome quality evaluation. The approach is attractive for scarce biological samples as it requires only a single tissue section. A limitation with RIN measured in bulk is that it provides an average of many spatially originating RIN values, whereas sRIN allows evaluation of each unique location. The principle of the molecular assay is simple—rRNA molecules are captured from the tissue section onto a uniformly covered rRNA capture slide, copied by reverse transcription and then hybridized with fluorescent probes along the complementary rRNA strand. The signal

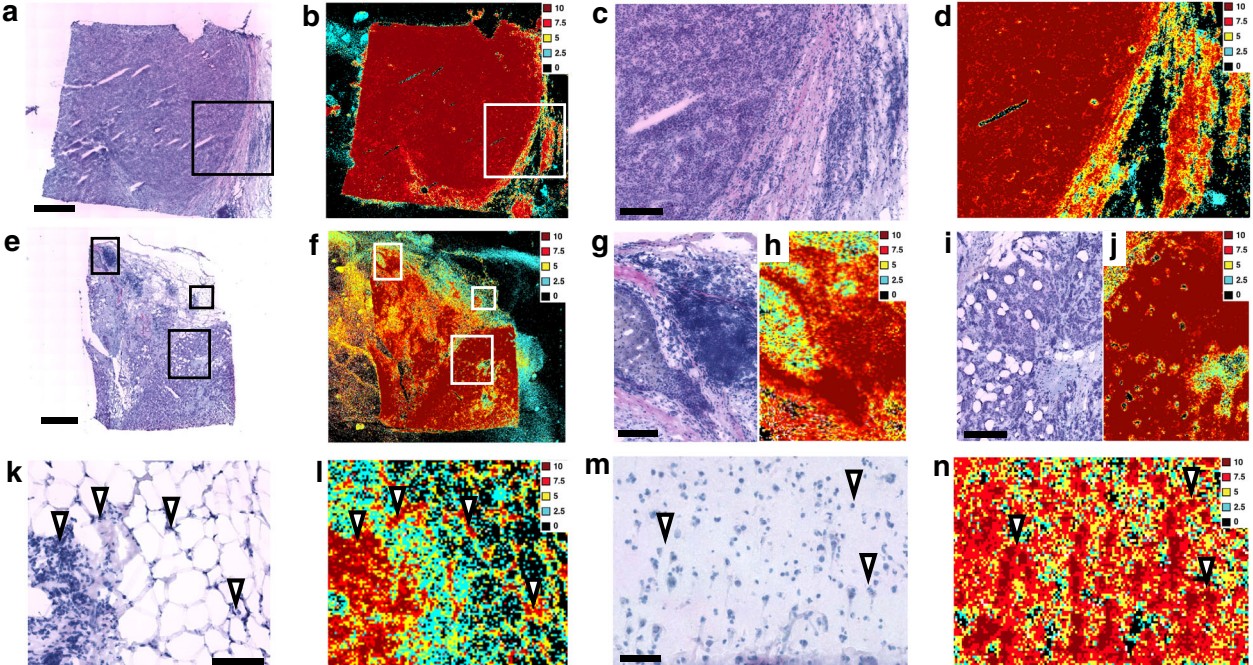

**Fig. 3 Application of the sRIN assay on well-preserved clinical samples. a–l** Luminal B breast tumor tissue sections from two patients ($n = 3$, technical replicates each). **a** HE image (scale bar: 1 mm) and **b** sRIN heat map, boxes enclose regions shown in close-up. Close-up of **c** HE image (scale bar: 200 μm) and **d** sRIN heat map. **e** HE image (scale bar: 1 mm) and **f** sRIN heat map, boxes enclose regions shown in close-up. Close-up of **g** HE image (scale bar: 150 μm) and **h** sRIN heat map. Close-up of **i** HE image (scale bar: 200 μm) and **j** sRIN heat map. Close-up of **k** HE image (scale bar: 100 μm) and **l** sRIN heat map, arrows show areas of very high (10) sRIN values and varied cell density. **m** HE image (scale bar: 80 μm) and **n** sRIN heat map of a gray matter region from a post-mortem brain ($n = 3$, biological replicates).

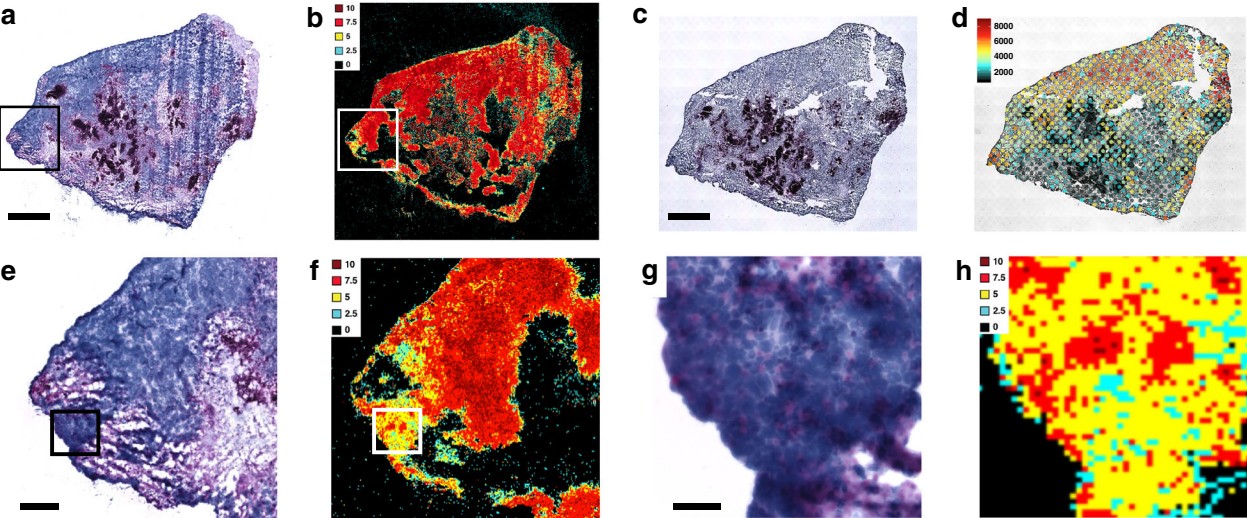

**Fig. 4 Specimen evaluation of spatial transcriptome quality with sRIN assay. a** HE image (scale bar: 1 mm) and **b** sRIN heat map of a malignant childhood brain tumor diagnosed as anaplastic medulloblastoma grade IV ($n = 4$, technical replicates), boxes enclose regions shown in close-up. **c** HE image (scale bar: 1 mm) and **d** ST analysis showing unique genes, of the same childhood brain tumor ($n = 2$, technical replicates). Close-up of **e** HE (scale bar: 200 μm) image and **f** sRIN heat map of the same childhood brain tumor where deep blue areas represent intact cell rich tumor and in light purple areas the tumor is necrotic. The sRIN heat map show areas of varied sRIN values in the tumor areas and no sRIN values in necrotic areas, boxes enclose regions shown in close-up. Close-up of **g** HE (scale bar: 40 μm) image and **h** sRIN heat map of the same childhood brain tumor show areas of similar HE staining pattern and varied sRIN values.

and the relative proportion from the probes will inform of the integrity of the rRNA in the tissue section.

Our data demonstrates that the RNA quality varies within tissue sections likely due to both technical and biological reasons. The resulting RNA integrity heat map provides us with a spatial RIN that can be used as a proxy of mRNA qualitative measurement in tissue specimens. Overall, we find it useful to analyze the sRIN heat map together with its corresponding HE image, to connect spatial RNA quality data to cells in different spatial context that may affect the desired outcome of a subsequent spatial gene expression analysis, e.g. freezing damages in tissue boundaries, hypoxic, necrotic regions, etc. Thus, the spatial

quality estimate can be used to guide decision making regarding sample selection before running a more costly spatial gene expression assay.

In our sRIN experiments on breast cancer specimens, representing well-preserved specimens, we found that high sRIN values were observed from areas with both high and low cell density supporting the use of the tissue for downstream spatial mRNA analysis. Furthermore, we applied the sRIN assay to GM and WM regions in human post-mortem brain. Here we observed high sRIN values (≥7.5) in cells from both regions while the cell drop-out, most likely occurring during the partitioning of cells in the cryosectioning step, was more frequent in WM. The GM regions of the human brain has been reported to have a more pronounced transcriptome activity compared to WM regions[15], thus a plausible explanation is that WM regions could be expected to be more sensitive to cell drop-out with the sRIN assay compared to GM regions. Practically, we can overcome this by allowing for a lower signal-to-noise threshold in the sRIN analysis and this can be recommended for cells where transcriptional activity is expected to be lower.

In general, areas containing cells with an even pattern of sRIN values ≥7.5, are considered being of sufficient quality to proceed with the more costly and time-consuming spatial analysis. In the analysis of the childhood brain tumor comparing sRIN and ST data it shows that areas of high-quality RNA yield high-quality gene expression detection. Importantly, distinct smaller areas of low-quality ST data were explained by corresponding low-quality RNA, which would not have been evident from a bulk analysis. In particular, we could observe areas of medium/low sRIN values (≤5) and also no signal from necrotic areas.

In summary, we demonstrate a simple spatial RNA integrity assay, sRIN, that can be used on a single tissue section. This in situ quality control screening has particular relevance to tissue samples that have been frozen[7,8,16,17] and/or in cases where specific spatial areas are of interest[6]. Thus, sRIN provides a new tool to assess widely used clinical and biobank tissue samples prior to more extensive spatial gene expression analysis.

## Methods

**Ethic statements**. Biospecimens were collected after written informed consent and regional ethics committees' approval of studies. Research was approved by the following ethical committees: etikprövningsnämnden in Stockholm, Stockholms Djurförsöksetiska Nämnd, etikprövningsnämnden in Lund and the Wales Research Ethics Committee.

**Design of primers and probes**. All primers and probes were designed using the primer3 software[18,19]. Sequence locations were picked for compatibility with both human (NR_003286.2) and mouse (NR_003278.3) 18S rRNA. Control probes were designed to include complementary sequences of three detection probes at a time with a 20 bases spacer sequence between each probe (Supplementary Table 1, Integrated DNA Technologies (IDT)). For all probes used during sequential hybridization to generate sRIN heat maps, a single fluorophore (Cy3) was selected to increase consistency and minimize variability in fluorescent signal between probe positions within the synthesized c18RNA.

**Printing of sRIN slides and control probe slides**. Oligonucleotides were immobilized on CodeLink-activated microscope glass slides (DN010025, Surmodics) according to manufacturer's instructions. All printed oligonucleotides had a 5′-end amino-C6 modification (Supplementary Table 1, IDT). A printing mixture was prepared according to Surmodics instructions for CodeLink activated slides, with final concentrations of 20 μM Capture probe or 10 μM of Control Probe and addition of 0.06% Sarkosyl (61747, Sigma-Aldrich) for even print patterns. The mixture (70 μl/well) containing a Capture probe for 18S rRNA was placed onto the CodeLink slide for one minute before removal by pipetting. Using a multi-well hybridization cassette (AHC1X16, ArrayIT Corporation) during printing of sRIN slides created multiple 7.5 × 7.5 mm fully coated capture areas. The mixture containing Control probe was instead added as 0.5 μl droplets to create four spots per capture area, then left to dry.

**Cell cultivation, frozen tissue sections, and RNA extraction**. The prostate cancer cell line LNCaP was obtained from Cell-Lines Service in Germany and was cultivated at 37 °C in a 5% $CO_2$ environment in media suggested by the provider, the cell line was not tested for mycoplasma and not authenticated. Cells were harvested during log-phase growth at 70–90% confluency. Fresh frozen tissues were embedded in OCT (4583, Tissue-Tek) and sections were collected during cryo-sectioning at −20 °C (CryoStar NX50 Cryostat), mixed with β-mercaptoethanol (444203, Calbiochem) in RLT lysis buffer (79216, Qiagen) at a 1:100 ratio, placed in Lysing Matrix D (116913050, MP Biomedicals), then lysed using FastPrep (MP Biomedicals, USA). For the post mortem brain specimen, GM and WM regions were separated using shallow indentations by a scalpel between regions prior to sectioning. Total RNA was extracted using RNeasy Plus Mini Kit (74134, Qiagen) according to manufacturer's instructions. Quality of the total RNA was determined using RNA 6000 Nano or Pico chip on the 2100 Bioanalyzer automated electro-phoresis system (Agilent Technologies Inc.), by calculating RIN. Concentration of the total RNA was determined using the Qubit RNA BR assay (Q10210, Life Technologies).

**Fragmentation of total RNA**. Universal human reference RNA (740000, Agilent Technologies) was prepared according to manufacturer's instructions. Fragmentation of total RNA was performed using NEBNext Magnesium RNA Fragmentation Module according to provided instructions (E6150S, New England Biolabs, NEB), at 94 °C for 10, 30 s, 2 or 4 min.

**Reverse transcription in solution for quantitative PCR**. Reverse transcription was performed in 20 μl reactions using 10 ng total RNA with a reaction mixture containing the following final concentrations; 1× First-strand buffer (included in, 18080085, Invitrogen), 5 mM DTT (included in, 18080085, Invitrogen), 1 M Betaine (61962, Sigma-Aldrich), 6 mM $MgCl_2$ (M8266, Sigma-Aldrich), 0.25 μM capture probe, 1 mM dNTPs (R0191, Thermo Fisher), 0.2 mg/ml BSA (B9000S, NEB), 10% DMSO (472301, Sigma-Aldrich), 20 U/μl SuperScript® III Reverse Transcriptase (18080085, Invitrogen), 2 U/μl RNaseOUT™ Recombinant Ribonuclease Inhibitor (10777019, Invitrogen). The reaction was performed at 42 °C for 50 min. The qPCR was performed using iQ SYBR Green supermix (1708880, Bio-Rad) for each 20 μl reaction using 1 ng of cDNA or 2 μl of released c18RNA footprint mixture or 2 μl DNase/RNase free water for negative controls, 0.5 μM forward and reverse primer, with the following heat cycling program: 3 min at 95 °C, 50 cycles of (10 s at 95 °C, 30 s at 55 °C, 30 s at 72 °C).

**Introducing in situ cross-links to the sRIN assay**. In experiments where cross-links were introduced a few key steps were performed differently. Fixation of tissue was done on the sRIN slide for 5 or 10 min at RT using PBS (09-9400, Medicago) containing 4% formaldehyde (F8775-25ML, Sigma-Aldrich), then washed briefly in PBS (09-9400, Medicago), dried for 1 min at 37 °C and then continued with propan-2-ol (A461-1, Fisher Scientific) treatment before HE staining. A tissue-specific (mouse olfactory bulb) permeabilization was performed before reverse transcription in situ. The permeabilization reactions were done, on the sRIN slide in a sealed hybridization cassette (AHC1X16, ArrayIT Corporation), first by adding a mixture of 1x Exonuclease I buffer (B0293S, NEB) and 0.2 mg/ml BSA (B9000S, NEB) for 30 min at 37 °C. Subsequently, tissue was briefly washed in 0.1 × SSC buffer (S6639, Sigma-Aldrich), treated with 0.1% pepsin (P7000, Sigma-Aldrich) dissolved in 0.1 M HCl (318965, Fluka) for 10 min at 37 °C, washed with 0.1 × SSC buffer (S6639, Sigma-Aldrich) and then reverse transcription in situ was performed immediately.

**Collection, sectioning, fixation, staining, and imaging**. Mouse olfactory bulbs from C57BL/6 mice (>2 months old, random genders) were collected immediately after euthanization and snap frozen in isopentane (03551-4, Fisher Scientific). Human breast cancer, childhood brain tumor, and post-mortem brain were snap-frozen in liquid nitrogen after collection. Tissues were embedded in cold OCT (Tissue-Tek, 4583), sectioned at 10 or 12 μm thickness at −20 °C (CryoStar NX50 Cryostat) and mounted onto an in-house printed sRIN slide with fully coated capture areas of 18S rRNA Capture probe. The rubber mask and hybridization cassette (AHC1X16, ArrayIT Corporation) was pre-cooled to −20 °C for at least 10 min. The sRIN slide with mounted tissue was first dried at 37 °C for 1 min, after which it was directly assembled into the pre-cooled hybridization cassette. Fixation of tissue was done using 100% pre-cooled −20 °C acetone (179124, Sigma-Aldrich) for 10 min, on ice (mouse olfactory bulb, childhood brain tumor, post-mortem brain) in a ventilated hood or at −20 °C (breast cancer). Then fixation solution was pipetted off, sRIN slide was incubated at 37 °C for 5 min, after which the rubber mask and hybridization cassette (AHC1X16, ArrayIT Corporation) were removed. Before staining, propan-2-ol (A461-1, Fisher Scientific) was added to the tissue on the sRIN slide and incubated at room temperature (RT) until dried. Then Mayer's Hematoxylin (S3309, Agilent) was added for 4 min (breast cancer) or 7 min (mouse, brain tumor and post-mortem brain), washed in water, incubated in Bluing buffer (CS702, Agilent) for 2 min, washed in water, stained with Eosin (HT110216, Sigma-Aldrich) in a Tris–acetic acid (Tris–AA) buffer at pH 6, ratio 1:20, for 20 s (mouse olfactory bulb, childhood brain tumor and post-mortem brain) or 60 s (breast cancer), then washed in water and finally dried at 37 °C for

5 min. Tris–AA buffer contained: 0.45 M Tris (T6791, Sigma-Aldrich), 0.5 M AA (242853, Sigma-Aldrich). After HE staining, sections were mounted with 85% glycerol (104094, Merck) and covered with a coverslip. Bright field imaging was done using the Metafer Slide Scanning Platform (Metasystems) where raw images were stitched together with the VSlide software (Metasystems). Glycerol was removed by holding the sRIN slide in water until the coverslip fell off. The sRIN slide was air dried until most remaining liquid evaporated and incubated at 37 °C for 2 min, after which the sRIN slide was again put into the hybridization cassette and reverse transcription in situ was then performed immediately.

**Reverse transcription in situ**. Reverse transcription in situ was performed, on the sRIN slide in a sealed hybridization cassette (AHC1X16, ArrayIT Corporation) by adding 70 µl reaction mixture per capture area, with final concentrations: 1× first-strand buffer (included in, 18080085, Invitrogen), 5 mM DTT (included in, 18080085, Invitrogen), 1 M betaine (61962, Sigma-Aldrich), 6 mM MgCl$_2$ (M8266, Sigma-Aldrich), 1 mM dNTPs (R0191, Thermo Fisher), 0.2 mg/ml BSA (B9000S, NEB), 50 ng/µl actinomycin D (A1410, Sigma-Aldrich), 10% DMSO (472301, Sigma-Aldrich), 20 U/µl SuperScript III Reverse Transcriptase (18080085, Invitrogen), 2 U/µl RNaseOUT Recombinant Ribonuclease Inhibitor (10777019, Invitrogen). The reaction was performed overnight at 42 °C, after which cDNA synthesis mixture was removed and the tissue washed with 0.1 × SSC buffer (S6639, Sigma-Aldrich).

**Removal of tissue and rRNA**. Tissue removal was performed, on the sRIN slide in a sealed hybridization cassette (AHC1X16, ArrayIT Corporation). For breast cancer, tissue was removed first by incubation with β-mercaptoethanol (444203, Calbiochem) in RLT lysis buffer (79216, Qiagen) at a 3:100 ratio at 56 °C for 1 h with continuous shaking at 300 rpm. All tissues were incubated with a 1:7 ratio of Proteinase K (19131, Qiagen) and PKD buffer (1034963, Qiagen) for 1 h at 56 °C using short intervals with gentle shaking at 300 rpm. After that the hybridization cassette (AHC1X16, ArrayIT Corporation) was removed and the sRIN slide washed under continuous shaking at 300 rpm as follows: first in 2 × SSC (S6639, Sigma-Aldrich) with 0.1% SDS (71736, Sigma-Aldrich) at 50 °C for 10 min, then in 0.2 × SSC (S6639, Sigma-Aldrich) at RT for 1 min and finally in 0.1 × SSC (S6639, Sigma-Aldrich) at RT for 1 min. The sRIN slide was then spin-dried and put back into the hybridization cassette. The rRNA was removed using a reaction mixture containing the following final concentrations: 1× first-strand buffer (included in, 18080085, Invitrogen), 0.4 mg/ml BSA (B9000S, NEB), and 16.3 mU/µl RNase H (18021071, Invitrogen). The reaction was performed for 1 h at 37 °C with gentle shaking at 300 rpm using short intervals and the sRIN slide capture areas were then washed with 0.1 × SSC buffer (S6639, Sigma-Aldrich), then treated with 60% DMSO (472301, Sigma-Aldrich) at RT for 5 min[20]. Finally, each capture area of the sRIN slide was washed three times with 0.1 × SSC buffer (S6639, Sigma-Aldrich).

**Hybridization and imaging of probes**. Probe hybridization to capture areas having a c18RNA footprint from 18S rRNA or printed control probe was done by pre-heating a hybridization mixture to 50 °C containing the following final concentrations at pH 8.0: 10 mM Tris–HCl (T6666, Sigma-Aldrich), 1 mM EDTA (108454, MERCK), 50 mM NaCl (S9888, Sigma-Aldrich), and 0.5 µM of fluorescently labeled probe (Supplementary 1, IDT) and adding this mixture to each capture area in the hybridization cassette (AHC1X16, ArrayIT Corporation) for 10 min at RT. Then the mixture was pipetted off and the sRIN slide was removed from the hybridization cassette (AHC1X16, ArrayIT Corporation) and washed under continuous shaking at 300 rpm first in 2 × SSC (S6639, Sigma-Aldrich) with 0.1% SDS (71736, Sigma-Aldrich) at 50 °C for 10 min, followed by 0.2 × SSC (S6639, Sigma-Aldrich) at RT for 1 min, then in 0.1 × SSC (S6639, Sigma-Aldrich) at RT for 1 min. Finally, the sRIN slide was spin-dried.

Imaging was done using a DNA microarray scanner (InnoScan 910, Innopsys, France) with the following settings: excitation wavelength 532 nm set to same gain for all images (gain range: 20–70) and 635 nm set to gain 1. Then images were analyzed for FU using (Mapix, Innopsys, France).

**De-hybridization of probes after an imaging step**. The capture areas were incubated with 60% DMSO (472301, Sigma-Aldrich) at RT for 5 min[20], after which they were washed three times with 0.1 × SSC buffer (S6639, Sigma-Aldrich). Important note: if the sRIN slide was stored dried prior to hybridization or de-hybridization, it was required to perform at short re-hydration step in 0.1 × SSC (S6639, Sigma-Aldrich) at RT for 5 min[20].

**Visualizing sRIN heat maps**. The tif images were aligned manually (Adobe Photoshop CC 2015), edges were cropped to make all tif images of the same size and then they were used to generate heat maps with the sRIN Script in RStudio. It is worth pointing out that when creating sRIN heat maps from tif images, the heat map resolution depends on the resolution of the tif file, which depends on the selected scanner settings used during imaging and the resolution limit of the scanner. The imaging was always started with the background fluorescence by scanning the slide before any labeled hybridization had been performed (P0). This

enabled visualization and subtraction of background noise from the sRIN data, which has proved particularly useful for samples where some tissue areas were not possible to remove from the slide, however such tissue areas should be excluded from analysis when interpreting a sRIN heat map.

The sRIN heat map script normalizes data pixel by pixel by creating a normalization dataset by subtracting the background auto-fluorescence (P0) and the signal-to-noise threshold from P1. The signal-to-noise threshold is calculated pixel by pixel based on a quantile, set by the user, from the recorded background auto-fluorescence (P0) from the sRIN slide before any probes are added. We recommend using the 0.75 quantile for most samples, if cells are expected to contain lower amounts of total RNA we recommend to set the signal-to-noise threshold to the 0.30 quantile. All probes (P1–P4) are normalized pixel by pixel by division with the normalization dataset and converted to sRIN scale, assembled and visualized as a sRIN heat map. Due to sometimes slight miss-alignments of pixels between tif files, resulting in false positive large sRIN values (>10), the sRIN script also produces a spatial miss-alignment plot of this error distribution. These miss-alignment errors should show an even spatial distribution, if not we recommend to re-do the manual image alignment of the tif files.

**Release of c18RNA footprint from the sRIN slide**. A total of 70 µl of release mixture was added to each capture area of the sRIN slide, with final concentrations: 1.1 × second strand buffer (10812014, Invitrogen), 0.088 mM dNTPs (R0191, Thermo Fisher), 0.2 mg/ml BSA (B9000S, NEB) and 0.1 U/µl USER enzyme (M5505L, NEB). The reaction was performed for 2 h at 37 °C with gentle shaking at 300 rpm using short intervals.

**ST in brief**. The ST library preparation protocol used has previously been described elsewhere[7,14]. Briefly, the method uses slides containing subarrays with spatially barcoded printed oligonucleotides with a 3′-end designed to capture poly-A tailed transcripts. Slides used here contained either 1007 unique spots of ~100 µm in diameter each (center-to-center spacing 200 µm) (SurModics)[7] or 1934 unique spots of ~110 µm in diameter each (center-to-center spacing 150 µm) (10× Genomics Inc). Tissue sections were placed on the printed subarray, fixated with formalin (10%), stained with HE, imaged at high resolution to capture tissue morphology. Then tissue was permeabilized, reverse transcription was performed overnight, enzymatic degradation and removal of tissue from the slide, cDNA was enzymatically cleaved from each subarray surface and collected separately into tubes ready for downstream library generation. The following sequential library preparation steps were conducted on a liquid-handling station (MBS 8000):[21] second-strand cDNA synthesis, in vitro transcription, adaptor ligation, and reverse transcription. Indexes and sequencing handles were added in a PCR, finished libraries purified and quantified. Libraries were sequenced on the Illumina NextSeq platform. Spot positions for individual subarrays were decoded by staining and imaging the remaining length of oligonucleotides printed in each spot after tissue and cDNA had been removed[14] and positions recorded from the image by using the ST Spot Detector[22].

**ST library preparation and sequencing**. Deviations from the detailed protocol[14] are described as follows: For one childhood brain tumor and two breast cancer specimens ST experiments were performed using 1007 unique barcoded spatial positions per subarray: Tissue was sectioned at 10 µm (brain tumor) or 16 µm (breast cancer) thickness at −20 °C (CryoStar NX50 Cryostat), 7 min (brain tumor), or 4 min (breast cancer) of Mayer's Hematoxylin (S3309, Agilent), 10 s (brain tumor) or 60 s (breast cancer) of Eosin (HT110216, Sigma-Aldrich) in a Tris–AA buffer at pH 6, ratio 1:20. The brain tumor was pre-permeabilized by an Exonuclease I buffer mixture with final concentrations: 1× Exonuclease I buffer (B0293S, NEB) and 0.2 mg/ml BSA (B9000S, NEB) at 37 °C for 30 min and the breast cancer by a collagenase mixture with final concentrations: 1× HBSS buffer (14025050, Thermo Fisher), collagenase I (17018029, Thermo Fisher), and 0.2 mg/ml BSA (B9000S, NEB) at 37 °C for 20 min. All were permeabilized by a pepsin mixture with final concentrations: 0.1 M HCl (318965, Fluka) and 0.1% pepsin (P7000-25G, Sigma-Aldrich) at 37 °C for 10 min. Tissue removal was performed using a two-step protocol[14] with following deviations: β-mercaptoethanol was prepared by adding 30 µl of β-mercaptoethanol (444203, Calbiochem) to 970 µl of Buffer RLT (79216, Qiagen) and the incubation time for the brain tumor was prolonged to 2 h in total. The Probe and cDNA release mix (step 42[14]) also included a final concentration of 0.2 mg/ml BSA (B9000S, NEB). Protocol steps from second-strand synthesis to cDNA purification (steps 53–122[14]) were performed using an automated approach described elsewhere[21].

For another childhood tumor ST experiment using 1934 unique barcoded spatial positions per subarray: tissue was sectioned at 12 µm thickness at −20 °C (CryoStar NX50 Cryostat), 7 min of Mayer's Hematoxylin (S3309, Agilent), 20 s of Eosin (HT110216, Sigma-Aldrich) in a Tris–AA buffer at pH 6, ratio 1:20. Pre-permeabilization by a collagenase mixture with final concentrations: 1× HBSS buffer (14025050, Thermo Fisher), collagenase I (17018029, Thermo Fisher), and 0.2 mg/ml BSA (B9000S, NEB) at 37 °C for 20 min. Permeabilization by a pepsin mixture with final concentrations: 0.1 M HCl (318965, Fluka) and 0.1% pepsin (P7000-25G, Sigma-Aldrich) at 37 °C for 10 min. Tissue removal was performed

using the one-step protocol[14]. The Probe and cDNA release mix (step 42[14]) also included a final concentration of 0.2 mg/ml BSA (B9000S, NEB). Protocol steps from second-strand synthesis to cDNA purification (steps 53–122[14]) were performed by using an automated approach described elsewhere[21], with a deviation (in step 86[14]) where only 1 µl of sample was used and diluted by adding 7 µl of DNase/RNase free water.

**Calculating hybridization efficiency of probes**. The FU was averaged within each sample (capture area) resulting in six estimates per probe.

**Comparison between sRIN and RIN**. The averaged sRIN was calculated for each sample ($n = 5$ biological replicates) by averaging the median sRIN value for all technical replicates. Then the Pearson correlation coefficient and $p$-value were computed between the paired RIN values and averaged sRIN values using the function cor.test from the R package stats, with 3 degrees of freedom.

**Comparison between 18S+ and 18S− total RNA with sRIN assay**. The fraction of pixels with a positive signal was calculated for each probe by dividing the number of pixels with a positive Cy3 signal by the number of pixels with a positive Cy3 signal for probe 1 (P1). Pixels with misaligned signals were not included in the calculation.

**Data processing, alignment, and annotation**. Sequencing data obtained with the ST method was processed using the ST Pipeline (v1.7.6, multi-mapping disabled)[23]. Raw sequencing reads were processed according to the following steps; sequencing reads were split into forward reads containing spatial barcode and UMI sequence and reverse reads containing the RNA sequence. Reverse reads were filtered from low read quality base pairs and the filtered sequencing reads were mapped against a reference genome (ENSEMBL genome assembly GRCh38, release 86) using STAR (v2.5.4a). Sequence alignments were then annotated using htseq-count (v0.10.0)[24] with gene annotations for the GRCh38 genome assembly (GENCODE release 25). Forward reads were demultiplexed by position from the unique spatial barcodes using the TaGD tool[25] and the spatial coordinates along with the UMI sequence were added to the sequencing alignment records of the corresponding read pair (reverse read). The annotated sequencing read alignments were grouped by spatial position (coordinate) and gene and within each gene/coordinate group, redundant alignments (PCR duplicates) were removed using the UMI sequences. Finally, the unique alignments were quantified for each gene/coordinate group to produce an alignment matrix with genes along the columns and coordinates along rows.

**Exon/intergenic reads ratio**. Raw reads were processed using the ST pipeline v1.7.6 with the following parameter settings; --htseq-no-ambiguous, --disable-multimap, --no-clean-up, --keep-discarded-files, --two-pass-mode, --htseq-mode union, --umi-quality-bases 6. The reference genome GRCh38 release 86 version 2 was used for mapping with STAR, and the GENCODE version 25 GTF file for the GRCh38 genome was used for annotation. Among the mapped reads, the total number of reads that were successfully annotated were used as an estimate for the total number of exonic reads and the discarded reads (i.e. reads that could not be annotated) were used as an estimate for intergenic (non-exonic) reads. The exon/intergenic mapping rate was then defined as the ratio between exonic reads and intergenic reads.

**Statistics and reproducibility**. If not otherwise specified, measurements were considered to be biological replicates when sampled from a different tissue specimen and/or region. If repeated measurements were performed on, e.g. adjacent/nearby tissue sections or from the same pool of total RNA they were considered to be technical replicates. For sRIN and ST experiments two to four replicates were used. Specimens included in the study were both from mouse and human, in total the sRIN assay was applied to seven distinct tissue regions. The sRIN assay was applied on both mouse and human specimens. We calculated the Pearson's correlation coefficient and $p$-value between sRIN and RIN using the function cor.test from the R package stats.

**Reporting summary**. Further information on research design is available in the Nature Research Reporting Summary linked to this article.

## Data availability
For source data underlying the graphs and charts, see Supplementary Data 1. For oligonucleotide sequences used in the sRIN assay, see Supplementary Information, Supplementary Table 1. The original gel files, HE images, tif files used to generate the sRIN heat maps and graphs, count matrixes and spot coordinates used for ST plots and exon/intergenic mapping rate data are available in a dataset on Mendeley[26]. Raw fastq files for the breast cancer and childhood brain tumor samples are available through a Materials transfer agreement with Å.B. (ake.borg@med.lu.se) and M.N. (monica. nister@ki.se), respectively, in line with GDPR regulations. Further data supporting the findings of this study are available from the corresponding author upon reasonable request.

## Code availability
The script for generating sRIN heat maps from tif files in RStudio, can be found at https://github.com/ludvigla/sRIN and on Mendeley[26]. For correct alignment of spot positions using Spatial Transcriptomics Spot Detector[22], the web-based tool was used and can be found at https://github.com/SpatialTranscriptomicsResearch/st_spot_detector. A detailed summary of the Spatial Transcriptomics Pipeline[23] used for processing of data can be found at https://github.com/SpatialTranscriptomicsResearch/st_pipeline.

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

## Acknowledgements

This work was supported by funds from the Swedish Childhood Cancer Foundation (Barncancerfonden), the Swedish Childhood Tumor Bank, the Swedish Cancer Society, the Swedish Research Council, the Swedish Foundation for Strategic Research. We would also like to acknowledge the Wellcome Trust (100308/Z/12/Z) the Medical Research Council (grant number MC_UU_12010/3), and the Oak Foundation (OCAY-15-520), the Multiple Sclerosis Tissue Bank funded by the Multiple Sclerosis Society of Great Britain and Northern Ireland, registered charity 207495 for providing post-mortem brain tissue and 10X Genomics Inc. We would like to thank Prof. Jonas Frisén for providing mouse samples for development of the method.

## Author contributions

L.K. participated in study design, analyzed data, and wrote the manuscript. K.C., L.L., E.G.V., A.St., L.S., A.M., M.Z., J.P.M., A.-L.S., participated in study design, performed experiments, analyzed data and contributed to writing the manuscript. E.B., A.Sh., G.P., participated in data analysis and contributed to writing the manuscript. Å.B., L.F., M.N., J.L., participated in study design and contributed to writing the manuscript. All authors read and approved the final manuscript.

## Funding

## Competing interests

The authors declare the following competing interests: L.K., K.C., L.L., E.G.V., L.S., A.M., and J.L., are scientific consultants for 10X Genomics Inc. All other authors declare no competing interests.
