## [Peer Review File · Communications Biology]

Reviewers' comments:

Reviewer #1 (Remarks to the Author):

In this manuscript, the authors develop a new assay to quantify the quality of RNA from tissue slices, analogous to an RNA integrity number used to bulk RNA analysis, but includes a spatial dimension. Assuming the sRIN method can be applied to tissue blocks before these blocks go through a spatial transcriptomic method and it is relatively inexpensive, the sRIN will be a useful tool for users to QC and select samples before proceeding with more expensive spatial transcriptomic experiments.

In sRIN, 18S rRNA is hybridized to a slide surface containing spots of capture probes specific for the 3' end of 18S rRNA. cDNA is made and then probes that span the 18S are hybridized and scanned in a sequential manner. Intact 18S and total RNA results in a relatively even signal across all probes, while degraded RNA would lead to a loss of signal from probes distal to the 3' end of the RNA. I only have some minor comments but feel this paper demonstrates a useful tool for groups doing ST methods whose usage will become more widespread.

1. The sentence in lines 63-65 is awkwardly worded. Maybe an alternative would be "We demonstrate its usability by applying sRIN to samples previously analyzed by spatial transcriptomics."
2. Please use the same Y-scaling for Figures S2 and S3
3. Line 99-101. I can't find the figure or table the sentence is referring to (Pearson correlation).
4. Was Probe 1 also used in Fig S6? Even with degraded RNA, one would expect to see an increase in signal in the 18S- as you go from Probe 4 over to Probe 1.
5. Line 141-142. Instead of referencing Table S2, did the authors mean to reference Fig 2?
6. It wasn't clear if there are discrete capture probe spots printed on the slide to make an array or if the entire slide surface is coated in solution. Assuming this is a true microarray, I don't think the resolution of the sRIN array or diameter of the spots is mentioned. Can the authors include this? If there aren't discrete spots, the authors shouldn't use the term microarray.
7. It wasn't clear how sRIN scores are calculated from the probe signals. Can the authors elaborate?
8. While it is nice to have a score analogous to the familiar RIN score, is there a simpler metric that can be used to determine the suitability of a sample for spatial transcriptomics? For instance, ST is a 3' based method so some fragmentation should be okay. Would the ratio of the P1 or P2 probe be a more accurate readout for screening samples than the sRIN score?
9. Fig S8 - Can the distances between the three slices be listed in the figure?
10. The strength of the method is the ability to screen tissues using a presumably lower cost and higher throughput assay (sRIN) before moving on to spatial transcriptomics methods which are more costly and lower throughput and to help authors analyze heterogeneous results within a single tissue. It would be useful for the authors to expand on how they see sRIN being used by researchers performing ST experiments. For example, could frozen tissue sections be screened for sRIN and depending on the results, only certain tissue blocks move onto a spatial transcriptomic method.

Reviewer #2 (Remarks to the Author):

Summary: Kvstad et al present an assay to quantify spatial RNA quality by creating an assay to profile 18S rRNA abundance in tissue sections. The authors apply this approach to several different datasets from mouse and human, including with complementary Spatial Transcriptomics (ST) gene expression data. While the approach seems potentially useful, it would have been helpful if the authors provided some guidelines for how to use their assay in practice. I also had some concerns related to a) how well 18S actually measures RNA quality and b) how 18S rRNA abundance associates with different cell types and overall transcriptional activity, described below.

Major:

- The authors should better describe how this spatial RNA Integrity number (sRIN) would be used in practice, for example, prior to a Spatial Transcriptomics (or more recent 10x Genomics Visium) experiment. How does one actually interpret the output of the sRIN assay to decide whether to

profile an adjacent tissue section with a much more expensive gene expression assay? There are clearly spatial patterns of sRIN (described more below), and as the authors demonstrate, average sRINs approximates regular RIN, so its unclear to me how and when I would actually use this assay prior to a spatial transcriptomics study. The authors should therefore provide some practical advice on using, interpreting and making decisions for running individual samples based on this sRIN assay at the end of their paper.

- The sRIN values seem to be very correlated to the cell density and perhaps cell types in Figure 1C. It's therefore unclear to me if some sort of normalization for the number of cells in each spot should be performed, as I would imagine there would be more absolute 18s rRNA from several cells compared to a single cell (or whether the 18s probe overlaps a cell or not). I think it would therefore be important for the authors to segment the histology images and plot the distribution of sRINs by cell density (or number of cells per spot...its not clear the spatial resolution and distribution of these 18s probes on the custom-printed slide). High correlation between cell density and sRIN might suggest this assay isn't really measuring RNA quality.

- Similarly, certain cell types like neurons might be more active than other cell types, and therefore have more rRNA. I think it would be important to assess this potential confounder using the paired Spatial Transcriptomics data by plotting the sRIN score for each ST spot versus different cell type marker gene expression levels in the brain dataset (like SNAP25, MBP, AQP4, PDGFRA, PLP1, CD74, etc) and confirm there is no correlation.

- Lastly, we have previously shown that while RIN might be a good measure of rRNA quality, it might not always be a good measure of total mRNA quality, particularly in brain tissue [PMID: 28634288]. There are several better measures of RNA quality that you can extract of RNA-seq data that can probably be analogously estimated in the ST data and then compared to the sRIN values. One measure relates to the "exonic mapping rate" ie what fraction of reads are assigned to genes during quantification. You could measure this at the ST spot/barcode level by removing duplicated reads via the UMI sequence, and then calculating the fraction of right/cDNA reads that mapped to genes. This is basically the ratio of deduplicated genic reads to total reads for each spatial barcode. The authors should plot this spot-level exonic mapping rate versus the rRIN values across each of the panel in Figure 1C, and Figure 2 as supplementary data. Another measure of RNA quality in brain tissue is the chrM mapping rate, which you could calculate at the spot level as well, which would just be the ratio of reads aligned to chrM for each spot/barcode. This chrM mapping rate should also be plotted against the sRIN values for each spot in Figures 1C and 2.

Andrew Jaffe (please leave my signature in)

Response to reviewers sRIN manuscript

We would like to sincerely thank the reviewers for their constructive evaluation and assessment of our manuscript. We would also like to thank for this opportunity to improve our work. We have now revised the manuscript based on the reviewers suggestions and questions and we have made changes addressed in the comments below.

Reviewers comments:

Reviewer #1 (Remarks to the Author):

In this manuscript, the authors develop a new assay to quantify the quality of RNA from tissue slices, analogous to an RNA integrity number used to bulk RNA analysis, but includes a spatial dimension. Assuming the sRIN method can be applied to tissue blocks before these blocks go through a spatial transcriptomic method and it is relatively inexpensive, the sRIN will be a useful tool for users to QC and select samples before proceeding with more expensive spatial transcriptomic experiments.

In sRIN, 18S rRNA is hybridized to a slide surface containing spots of capture probes specific for the 3' end of 18S rRNA. cDNA is made and then probes that span the 18S are hybridized and scanned in a sequential manner. Intact 18S and total RNA results in a relatively even signal across all probes, while degraded RNA would lead to a loss of signal from probes distal to the 3' end of the RNA. I only have some minor comments but feel this paper demonstrates a useful tool for groups doing ST methods whose usage will become more widespread.

We would like to thank the reviewer for the support of our work and valuable points for improving upon it.

1. The sentence in lines 63-65 is awkwardly worded. Maybe an alternative would be “We demonstrate its usability by applying sRIN to samples previously analyzed by spatial transcriptomics.

We have updated the manuscript text to address this suggestion.

2. Please use the same Y-scaling for Figures S2 and S3

We agree that this change will improve interpretability of the plots and we have now updated figures S2 and S3.

3. Line 99-101. I can't find the figure or table the sentence is referring to (Pearson correlation).

We thank the reviewer for pointing this out and we have updated the text to clarify this sentence. We also included a new figure in supplementary displaying the data used to calculate the Pearson correlation.

4. Was Probe 1 also used in Fig S6? Even with degraded RNA, one would expect to see an increase in signal in the 18S- as you go from Probe 4 over to Probe 1.

We thank the reviewer for bringing this up for discussion.

Probe 1 was used to normalize the values for probes 2-4 and thus is not displayed. We have clarified this in the figure text.

Yes, we agree that one would expect to see an increase in signal in the 18S- data as one goes from Probe 4 to Probe1. We do see such a pattern, which is more clearly displayed when separating the data and allowing for different y-axis scales. We have included such a figure here below in our response. For the manuscript we have instead selected to use the same y-axis scale in the figure, to highlight and display the difference in signal from samples with (18S+) and without (18S-) the 18S rRNA peak.

5. Line 141-142. Instead of referencing Table S2, did the authors mean to reference Fig 2?

We thank the reviewer for pointing this out. We agree that this reference could be more clear to the reader and have updated the text accordingly.

6. It wasn't clear if there are discrete capture probe spots printed on the slide to make an array or if the entire slide surface is coated in solution. Assuming this is a true microarray, I don't think the resolution of the sRIN array or diameter of the spots is mentioned. Can the authors include this? If there aren't discrete spots, the authors shouldn't use the term microarray.

We thank the reviewer for bringing this up for discussion. The entire slide surface is coated in solution. We have reconsidered our naming choice and updated the text accordingly, now referring to it as a 'sRIN slide' with 'fully coated capture areas'.

7. It wasn't clear how sRIN scores are calculated from the probe signals. Can the authors elaborate?

We thank the reviewer for this suggestion. We have included a new section in material and methods, 'Visualizing sRIN heat maps', here describing how the sRIN score is calculated using the sRIN script.

8. While it is nice to have a score analogous to the familiar RIN score, is there a simpler metric that can be used to determine the suitability of a sample for spatial transcriptomics? For instance, ST is a 3' based method so some fragmentation should be okay. Would the ratio of the P1 or P2 probe be a more accurate readout for screening samples than the sRIN score?

We thank the reviewer for this suggestion. We agree with the reviewer that it is nice to have a score that is analogous to the familiar RIN score and this is our main reason for selecting this score and scale. In this study we did select one spatial transcriptomics method to use for validation that is 3' based, but there are other alternative spatial methods. For example, Visium from 10X genomics, generates full length cDNA libraries as part of the library preparation protocol and would benefit to score the full length of rRNA.

9. Fig S8 - Can the distances between the three slices be listed in the figure?

We thank the reviewer for this suggestion and have updated the figure including estimated distances between the slices.

10. The strength of the method is the ability to screen tissues using a presumably lower cost and higher throughput assay (sRIN) before moving on to spatial transcriptomics methods which are more costly and lower throughput and to help authors analyze heterogeneous results within a single tissue. It would be useful for the authors to expand on how they see sRIN being used by researchers performing ST experiments. For example, could frozen tissue sections be screened for sRIN and depending on the results, only certain tissue blocks move onto a spatial transcriptomic method.

We thank the reviewer for this suggestion, we have made some major restructuring and updating of the text, mainly in Results and Discussion to expand on how to use and interpret the sRIN assay.

Reviewer #2 (Remarks to the Author):

Summary: Kvastad et al present an assay to quantify spatial RNA quality by creating an assay to profile 18S rRNA abundance in tissue sections. The authors apply this approach to several different datasets from mouse and human, including with complementary Spatial

Transcriptomics (ST) gene expression data. While the approach seems potentially useful, it would have been helpful if the authors provided some guidelines for how to use their assay in practice. I also had some concerns related to a) how well 18S actually measures RNA quality and b) how 18S rRNA abundance associates with different cell types and overall transcriptional activity, described below.

We thank the reviewer for the interest in our work and valuable points for improving upon it.

Major:

- The authors should better describe how this spatial RNA Integrity number (sRIN) would be used in practice, for example, prior to a Spatial Transcriptomics (or more recent 10x Genomics Visium) experiment. How does one actually interpret the output of the sRIN assay to decide whether to profile an adjacent tissue section with a much more expensive gene expression assay? There are clearly spatial patterns of sRIN (described more below), and as the authors demonstrate, average sRINs approximates regular RIN, so its unclear to me how and when I would actually use this assay prior to a spatial transcriptomics study. The authors should therefore provide some practical advice on using, interpreting and making decisions for running individual samples based on this sRIN assay at the end of their paper.

We thank the reviewer for these suggestions and discussion points. We agree that we could expand more upon how to use and interpret sRIN. We have made major updates and reconstruction of the text to facilitate this, mainly in the Results and Discussion. Taking into account the reviewer's suggestion we have included practical advice on how to use, interpret and make decisions - based on the sRIN data – on whether and how to proceed with a sample for further spatial transcriptomics analysis.

- The sRIN values seem to be very correlated to the cell density and perhaps cell types in Figure 1C. It's therefore unclear to me if some sort of normalization for the number of cells in each spot should be performed, as I would imagine there would be more absolute 18s rRNA from several cells compared to a single cell (or whether the 18s probe overlaps a cell or not). I think it would therefore be important for the authors to segment the histology images and plot the distribution of sRINs by cell density (or number of cells per spot...its not clear the spatial resolution and distribution of these 18s probes on the custom-printed slide). High correlation between cell density and sRIN might suggest this assay isn't really measuring RNA quality.

We thank the reviewer for raising this point for discussion. We have realized that our choice in naming it a 'microarray' was a very poor one indeed. The custom printed slide is an array of fully coated capture areas, thus for the sRIN assay we do not have spots - only for the Spatial Transcriptomics (ST) data. We apologize for this and have updated the naming, now referring to it as a 'sRIN slide' with 'fully coated capture areas'. With that in mind, we thank the reviewer again for raising important discussions regarding correlations between cell density and sRIN. To address this we have updated the text to more clearly explain that the sRIN assay uses fully coated capture areas and single cell layer sections. We have also included in Figure 2C a close-up on an area with high sRIN values from areas with varied cell density, to visualize that cell density does not correlate with sRIN. We have also updated

Figure 1, including a close-up image and expanded on this in the text to make it more clear to the reader how to interpret the image and data. E.g. we have added text explaining that we expect some cell-drop out in the sRIN data due to how some cells are partitioned in the tissue. And while cell drop-out can occur, for areas of high RNA integrity it is also expected to observe cells with high sRIN values.

- Similarly, certain cell types like neurons might be more active than other cell types, and therefore have more rRNA. I think it would be important to assess this potential confounder using the paired Spatial Transcriptomics data by plotting the sRIN score for each ST spot versus different cell type marker gene expression levels in the brain dataset (like SNAP25, MBP, AQP4, PDGFRA, PLP1, CD74, etc) and confirm there is no correlation.

We thank the reviewer for raising this point for discussion. We agree that this is a potential confounder and important to address. Although we acknowledge the reviewers suggestions for addressing this, we unfortunately do no longer have access to the human brain tissue specimens. We do however have previously generated sRIN data and RIN data for both grey matter (GM) and white matter (WM) areas, which we have included in the new supplementary figure 8. We thank the reviewer for pointing out that certain cell types could have varying gene activity and contain more rRNA than others. We have addressed this by including a source from previous published literature, in the discussion, where the authors reported that GM regions from their human brain specimens had a more pronounced transcriptome activity than the WM regions¹. This, taken together with our own data where we measure bulk RIN from both GM and WM separately and performed sRIN on close by sections, show that we can indeed still record clear sRIN values of high signal from cells in both GM and WM. We do acknowledge that there is a clear difference in the visual representation of sRIN between GM and WM, this we attribute to a higher expected cell drop-out rate in WM, due to an expected lower transcriptional activity. We also make specific recommendations to the reader on what settings to use in the sRIN analysis for cells with expected lower transcriptional activity in the new material and methods part 'Visualizing sRIN heat maps'. Keeping in mind that bulk analysis of total RNA for obtaining a RIN value has a qualitative lower limit of 50 pg, which is a bit more total RNA than what is expected from one single cell, the sRIN assay can in fact make a qualitative measure from single cells *in situ*. From our point of view, we feel this addresses the potential confounder of applying sRIN for cells with expected lower transcriptional activity.

- Lastly, we have previously shown that while RIN might be a good measure of rRNA quality, it might not always be a good measure of total mRNA quality, particularly in brain tissue [PMID: 28634288]. There are several better measures of RNA quality that you can extract of RNA-seq data that can probably be analogously estimated in the ST data and then compared to the sRIN values. One measure relates to the "exonic mapping rate" ie what fraction of reads are assigned to genes during quantification. You could measure this at the ST spot/barcode level by removing duplicated reads via the UMI sequence, and then calculating the fraction of right/cDNA reads that mapped to genes. This is basically the ratio of deduplicated genic reads to total reads for each spatial barcode. The authors should plot this spot-level exonic mapping rate versus the rRIN values across each of the panel in Figure 1C, and Figure 2 as supplementary data. Another measure of RNA quality in brain tissue is the chrM mapping rate, which you could calculate at the spot level as well, which would just be

the ratio of reads aligned to chrM for each spot/barcode. This chrM mapping rate should also be plotted against the sRIN values for each spot in Figures 1C and 2.

We thank the reviewer for raising this point for discussion. While we do agree that there are limitations to using RIN as an estimate of degradation in the mRNA population, as pointed out by the reviewer and other studies on transcript specific decay rates^{2,3}, which highlight that RIN is just a rough guide to determine the overall RNA integrity of a sample. We would expect to find more differences in RNA quality if performing a study on individual mRNA transcripts, however spatial transcriptomics data today is too sparse for such in-depth analysis to be performed and would be something to measure after an experiment has already been done. With the sRIN assay we instead present a more global tissue section analysis of spatial RNA quality before any spatial data has been generated. Although we acknowledge the reviewers suggestions for addressing this using spatial transcriptomics data, we unfortunately do no longer have access to the human brain tissue specimens or the specimen in Figure 1. We have however been able to obtain spatial transcriptomics data for the breast cancer specimens and have performed an 'exonic mapping rate' analysis, adapted to the current spatial transcriptomics pipeline⁴ available, for these and the childhood brain tumor. Due to limitation with the ST pipeline, we had to include PCR duplicates in the analysis. The ST pipeline can output annotated and discarded reads but only before removal of PCR duplicates. However, we would argue that it is a reasonable assumption that the PCR duplication level is the same for exonic and intergenic reads and therefore the estimated ratio that we present here as 'exon/intergenic mapping rate' should work as a proxy for the ratio calculated without PCR duplicates.

Reference

1. Mills, J. D. *et al.* Unique Transcriptome Patterns of the White and Grey Matter Corroborate Structural and Functional Heterogeneity in the Human Frontal Lobe. *PLoS One* **8**, 1–18 (2013).
2. Wang, Y. *et al.* Precision and functional specificity in mRNA decay. *Proc. Natl. Acad. Sci. U. S. A.* **99**, 5860–5865 (2002).
3. Yang, E. *et al.* Decay rates of human mRNAs: correlation with functional characteristics and sequence attributes. *Genome Res.* **13**, 1863–1872 (2003).
4. Navarro, J. F., Sjostrand, J., Salmen, F., Lundeberg, J. & Stahl, P. L. ST Pipeline: an automated pipeline for spatial mapping of unique transcripts. *Bioinformatics* **33**, 2591–2593 (2017).

REVIEWERS' COMMENTS:

Reviewer #1 (Remarks to the Author):

After reviewing the rebuttal and the changes to the manuscript and supplemental material, I am satisfied with the author's response.

Reviewer #2 (Remarks to the Author):

The authors addressed my concerns